# CSN5A Subunit of COP9 Signalosome Temporally Buffers Response to Heat in Arabidopsis

**DOI:** 10.3390/biom9120805

**Published:** 2019-11-29

**Authors:** Amit Kumar Singh, Brijesh Singh Yadav, Shanmuhapreya Dhanapal, Mark Berliner, Alin Finkelshtein, Daniel A. Chamovitz

**Affiliations:** 1School of Plant Sciences and Food Security, Tel Aviv University, Tel Aviv 6997801, Israel; amit.mku@gmail.com (A.K.S.); brijeshbioinfo@gmail.com (B.S.Y.); shanmuhapreya@gmail.com (S.D.); berliner.mark@gmail.com (M.B.); alinfin@gmail.com (A.F.); 2The Jacob Blaustein Institutes for Desert Research, Ben-Gurion University of the Negev, Midreshet Ben-Gurion 8499000, Israel; 3Department of Bioengineering, University of Information Science and Technology (UIST) St. Paul 6000, Republic of Macedonia

**Keywords:** COP9 signalosome, hypomorphic mutants, cullin deneddylation, auxin signaling, VENUS reporter construct

## Abstract

The COP9 (constitutive photomorphogenesis 9) signalosome (CSN) is an evolutionarily conserved protein complex which regulates various growth and developmental processes. However, the role of CSN during environmental stress is largely unknown. Using Arabidopsis as model organism, we used CSN hypomorphic mutants to study the role of the CSN in plant responses to environmental stress and found that heat stress specifically enhanced the growth of *csn5a-1* but not the growth of other hypomorphic photomorphogenesis mutants tested. Following heat stress, *csn5a-1* exhibits an increase in cell size, ploidy, photosynthetic activity, and number of lateral roots and an upregulation of genes connected to the auxin response. Immunoblot analysis revealed an increase in deneddylation of CUL1 but not CUL3 following heat stress in *csn5a-1*, implicating improved CUL1 activity as a basis for the improved growth of *csn5a-1* following heat stress. Studies using DR5::N7-VENUS and DII-VENUS reporter constructs confirm that the heat-induced growth is due to an increase in auxin signaling. Our results indicate that CSN5A has a specific role in deneddylation of CUL1 and that CSN5A is required for the recovery of AUX/IAA repressor levels following recurrent heat stress to regulate auxin homeostasis in Arabidopsis.

## 1. Introduction

The COP9 (constitutive photomorphogenesis 9) signalosome (CSN) is an evolutionarily conserved multiprotein complex comprised of eight subunits (CSN1 to CSN8) [1]. CSN was discovered in Arabidopsis as a repressor of light-regulated development [2,3]. Complete loss of any subunit results in pleiotropic *constitutively photomorphogenic/de-etolated/fusca (cop/det/fus)* phenotype with characteristic stunted growth, open cotyledons in dark-grown seedling, short hypocotyl, and anthocyanin pigment accumulation [3,4].

Other than photomorphogenesis, the CSN regulates a number of hormonal signaling pathways through its action as a deneddylase regulating ubiquitin-mediated protein stability [5]. CSN regulates responses to auxin, jasmonate, and gibberellic acid, as well as flower development, through its regulation of Cullin-RING ubiquitin E3 ligases (CRLs) such as the SKP1, Cullin and F-box-containing protein (SCF) complexes SCF^TIR1^, SCF^COI1^, SCF^SLI1^, SCF^CFK1^, and SCF^UFO^ [6,7,8,9,10]. The CSN also regulates other CRLs such as those containing CUL3 (cullin 3) and CUL4 [11,12].

CSN plays a critical role in protecting plants from biotic stress by regulating N gene-mediating resistance to tobacco mosaic virus [13] and jasmonic acid-dependent plant defense response [14]. CSN is also involved in double-stranded break repair [15] and nucleotide excision repair [16]. *csn* mutants show auto-degradation of a CRL substrate receptor, which is regulated in cell-type-specific manner [17].

CSN is not only involved in developmental processes [11] but also plays significant role in cell cycle progression [18]. *csn* mutants show delay in S-phase progression in yeast [19], defective S phase progression in mouse thymocytes [20], and G2 phase arrest in Arabidopsis roots [18].

Enzymatically, CSN is a metalloprotease which cleaves neural precursor cell expressed, developmentally downregulated 8 (NEDD8) from the cullin subunit of CRLs by a process called deneddylation [7,21]. This catalytic activity is located in the JAB1/MPN/Mov34 metalloenzyme (JAMM) motif of CSN5 subunit [22].

Complete loss of any CSN subunit leads to seedling lethality in early stage, which obstructs further analysis of the role of subunits in plant growth and development [4]. However, in Arabidopsis, subunit CSN5 is encoded by two partially redundant genes [23], which allows mutant plants to grow to adulthood [24]. Subsequent discovery of viable hypomorphic mutants of other subunits enabled to study the role of CSN subunits in the adult and reproductive stages. These hypomorphic mutants can be broadly classified into two categories: 1. Mutants compromised in cullin deneddylation and auxin/3-indoleacetic acid (AUX/IAA) degradation (e.g., *csn1-10*, *csn2-5*, *csn4-2035*, *csn5a-1*, and *csn5a-2*) and 2. Mutants with normal cullin deneddylation and AUX/IAA degradation activity (e.g., *csn3-3*, *csn5b-1*, *csn6a-1*, and *csn6b-1*) [5]. *csn5a-1* exhibits hyper-neddylation of CUL1, CUL3, and CUL4, whereas *csn5b-1* shows normal cullin neddylation similar to wild type [25]. Studies have shown that, while CSN5A is crucial for seed germination, CSN1 plays a prominent role in seed maturation. The seed germination phenotype of *csn1-10* is due to over-accumulation of RGL2; however, the germination phenotype of *csn5a-1* is not only because of RGL2 but also ABI5. Thus, ABI5 is especially affected in *csn5a-1* but not in *csn1-10* [26].

In this study, we employed viable hypomorphic *csn* mutants to study the role of CSN in response to abiotic stress. We found that, while these mutants are hypersensitive to salinity and UV-C, growth of *csn5a-1* was enhanced after heat stress. This enhanced growth is likely a result of numerous parameters including increased photosynthetic output and increase in CUL1 deneddylation and auxin activity. Thus, CSN5A is required to buffer plants during heat by maintaining auxin homeostasis.

## 2. Materials and Methods

### 2.1. Plant Material and Growth Conditions

All the Arabidopsis lines used in this work were of Columbia-0 (Col-0) background. The transgenic lines were described earlier: *csn5a-1* and *csn5b-1* [24], *csn1-10* [12], *spa1-3-4* [27], *cop1-4* [28], DR5::N7-VENUS [29], and DII-VENUS [30]. Sterile seeds were sown on petri plates containing 1× Murashige and Skoog salts (MS) [31], 0.8% agar, 1% sucrose, and 0.05% MES (2-(N-morpholino) ethanesulfonic acid) at pH 5.7. After 2 days of cold stratification (4 °C in dark) plates were transferred to the growth chamber at 21 °C under long day condition (16 h white light at 100 µmol m^2^s^−1^ and 8 h darkness) at 70% relative humidity; 10 days after sowing (DAS), seedlings were transferred to the soil, and stress treatment was given at 14 DAS. For root and confocal microscopy studies, seedlings were grown in liquid MS.

### 2.2. Stress Treatments

Fourteen DAS, seedlings in soil were treated with either 44 °C or 28 °C for 2 h a day, starting at 11:00, for 7 d. Relative humidity of the chamber was maintained at 55–70% during heat treatment. Other conditions assayed starting at 14 DAS include 200 mM NaCl for 21 d, drought (21 d), UV-C (5000 erg, 1 min), flood (7 d), and cold (4 °C overnight, 7 d). Phenotypes were analyzed either 30, 60 or 70 DAS according to the experiment. For lateral root study, seedlings 10 DAS were grown in liquid MS and treated with heat (2 h, 44 °C, 7 d) and lateral roots were analyzed at 17 DAS.

### 2.3. Phenomics

Rosette area and photosynthetic parameters were examined by Plant Screen^TM^ Phenotyping System (Photon Systems Instruments (PSI), Drasov, Czech Republic). Seedlings 10 DAS grown in MS were transferred to PSI standard pots (one seedling per pot), and imaging was done on control and heat-treated seedlings at 14, 18, 21, 25, 28, 32 and 35 DAS.

### 2.4. Analysis of Cell Size, Cell Number and Stomatal Density

Cell size and cell number studies were performed as described in literature [32]. Sixth true leaf of 30 DAS seedlings was fixed, and chlorophyll was cleared with 70% ethanol. Images of cells were taken under differential interference contrast (DIC) of an Olympus DP71 microscope. Cell size and cell number were calculated using ImageJ software (http://imagej.-nih.gov/ij/) similar to Reference [33].

### 2.5. Ploidy Analysis

Ploidy analysis was performed as described in literature [34] protocol with slight modification; 20 mg of seedling at 30 DAS control and heat-treated were taken in test tubes containing 1 mL of ice cold Galbraith’s buffer (45 mM MgCl_2_, 20 mM MOPS (3-(N-morpholino)propanesulfonic acid), 30 mM sodium citrate, and 0.1% TritonX-100 at pH 7) and lysed by tissue lyser (1.5 min at 25 Hz). Homogenate was filtered through 40 µm nylon mesh. The filtrate was centrifuged at 200 *g* for 5 min to sediment the nuclei, and the pellet was suspended in 200 µL of Galbraith’s buffer. The samples were stained with 50 µg/mL propidium iodide containing 50 µg/mL RNase. After staining, samples were analyzed using Stratedigm S1000Exi flow cytometer (San Jose, CA, USA). Data analysis was done by FlowJo software (Version10) (Tree Star, Inc. Ashland, OR, USA).

### 2.6. qRT-PCR Analysis

RNA was extracted from 21-DAS control and heat-treated seedlings 3 h after the treatment using TriReagent (MRC) method according to the manufacture instruction. cDNA was synthesized using qScript cDNA Synthetic Kit (Quantabio, Beverly, MA, USA), and qRT-PCR was performed using PerfectCT SYBR Green FastMix. Actin was used as a normalization gene. The experiments were performed in three to five biological replicates with three technical repetitions for each biological sample. List of primers and genes used for qRT-PCR is given in Appendix A. For statistical analysis, unpaired Student’s test was used.

### 2.7. Western Blot

For immunoblots, proteins were extracted 3 h following heat treatment (2 h, 44 °C, 14–21 DAS) from heat-treated and control seedlings. Seedlings were frozen in liquid nitrogen, homogenized in extraction buffer (50 mM Tris-HCl, pH 7.4, 150 mM NaCl, 10 mM MgCl_2_, 5 mM EDTA (ethylenediaminetetraacetic acid), 5% glycerol, 1 mM PMSF (phenylmethylsulfonyl fluoride), 1 mM DTT (dithiothreitol), and 1X complete cocktail (Roche 1836145)), and centrifuged at 14,000 rpm for 30 min at 4 °C. Protein concentration was determined by Bradford protein assay, and equal amounts of protein were loaded for western blot with specific antibodies anti-CSN5 (Abcam, ab195635, Cambridge, UK), anti-CUL3 (Enzo, BML-PW0470, Enzo Life Sciences, NY, USA), and anti-CUL1 (Enzo, BML-PW0190).

### 2.8. Confocal Microscopy

Seedlings 6 DAS, from the DR5::N7-VENUS and DII-VENUS lines, grown on liquid MS were treated with heat (2 h, 44 °C, 7 d); roots were stained with propidium iodide (2 µg/mL) and imaged using Zeiss LSM780 confocal microscope 3 h and 3 d after the heat treatment. For the DR5::N7-VENUS line, fluorescence emission was collected between 520–570 nm (band pass) for yellow fluorescent protein (YFP). For the DII-VENUS line, confocal imaging was done with a ChS1 detector for YFP signals, and the fluorescence emission was collected between 508–543 nm (band pass). For propidium iodide, fluorescence emission was collected between 595–695 nm (band pass). YFP fluorescence was quantified using ImageJ software.

## 3. Results

### 3.1. Heat Treatment (44 °C, 2 h, 7 d) Enhances the Growth and Lateral Roots of csn5a-1

To study the role of CSN in mediating responses to environmental stress, we studied two hypomorphic mutants, *csn1-10* and *csn5a-1*, under various growth conditions as detailed in the Methods section. We chose these strains since, opposed to loss-of-function *csn* mutants which are seedling lethal, these strains are viable, reaching seed sets.

At the stage of adult plants, both mutants and WT (wild type) showed slower growth after salt and drought stress, though *csn5a-1* was comparatively less affected than *csn1-10* or the WT, with *csn5a-1* leaves remaining green whereas the leaves of WT and *csn1-10* were mostly dried (Figure 1a–c). UV-C reduces the growth of all plants, with *csn5a-1* and *csn1-10* being hyper-sensitive. Flood and cold stress in the conditions assayed did not show any prominent effect on the growth of WT, *csn1-10*, as well as *csn5a-1* (Figure 1d–f).

Most interestingly, the growth of *csn5a-1* was enhanced after the transient heat regimen of two hours a day at 44 °C for seven days (Figure 1c), whereas these conditions did not affect the growth of WT or *csn1-10*. Similar experiments but with a milder heat stress of two hours a day at 28 °C showed that the growth of *csn5a-1* increased linearly with the highest growth observed in *csn5a-1* leaf rosette area at 44 °C (Figure 1g).

To assay if the heat stress-induced growth is specific to *csn5a-1* or common for other photo-morphogenesis mutants, we studied additional CSN hypomorphic mutants (*csn5b-1* and *csn1-10*), as well as *cop1-4* and a SPA (suppressor of phyA-105) triple mutant (*spa1*, *spa3* and *spa4*). We observed a significant increase in the growth of *csn5a-1* after heat stress, with the leaf rosette area doubling by 28 DAS (Appendix A). However, this same heat stress did not affect growth of Col-0 (Appendix A) or any of the other mutants assayed (Appendix A), indicating the specificity of heat stress-induced growth for *csn5a-1.* Further, we observed an increase in the number of lateral roots following heat stress in *csn5a-1* mutants grown in liquid media (Appendix A).

### 3.2. Enhanced Growth of csn5a-1 after Heat Treatment Correlates with an Increase in Photosynthetic Capacity

To understand how the heat stress may change physiology, leading to the enhanced growth of *csn5a-1*, we measured several photosynthetic parameters using a high-throughput phenotyping robot. We collected data for chlorophyll fluorescence decline ratio in steady state (Rfd_Lss), which is an indicator of potential photosynthetic activity of leaf and directly correlates to net photosynthetic CO_2_ assimilation rate [35]. We detected a significant increase in Rfd_Lss after 7 d heat stress in *csn5a-1*, which remains high up to 11 d following heat stress before declining to the level of *csn5a-1* control (Figure 2a). However, the increased Rfd_Lss in Col-0 (Figure 2b), *csn1-10* (Figure 2c), and *csn5b-1* (Figure 2d) disappeared by the fourth day following the last heat stress. This result led us to suggest that CSN5A provides buffering activity during heat stress, as it appears necessary for plants to return to ground photosynthetic rates following the stress. Rfd_Lss did not change after heat stress in *cop1-4* and *spa1-3-4*, further suggesting that the heat response detected is specific for the mutation in CSN5A and not a general response of perturbation in photomorphogenesis (Figure 2e,f). Further, we observed a different pattern of leaf greenness in WT and *csn5a-1* mutant following the heat treatment (Appendix A), where the variation in relative hue abundance indicates the change in the level of chlorophyll content after heat treatment. The hue for pale green (RGB 90, 98, 58) changes more dramatically in *csn5a-1* after 7 d of heat treatment. The percentage of pale green increased almost two-fold (58% in *csn5a-1* heat versus 30% in *csn5a-1* control) following heat treatment of *csn5a-1* mutant in 21-DAS seedling. However, such drastic change in hue abundance was not observed in WT after heat treatment. Change in the hue abundance in *csn5a-1* mutants after heat treatment suggests a change in chlorophyll content, which improves photosynthesis. In summary, heat stress leads to an increase in photosynthetic activity, which increases net photosynthetic CO_2_ assimilation rate in *csn5a-1* and correlates with an increase in plant size.

### 3.3. Heat Treatment Increases the Cell Size and Ploidy of csn5a-1

To further study the influence of heat stress leading to the increased growth in *csn5a-1,* we measured cell size and cell numbers on the abaxial side of the 6th true leaf of seedlings 30 DAS. Differential interference contrast (DIC) images showed an increase in the abaxial epidermal cell size of *csn5a-1* after heat stress, whereas Col-0 and *csn1-10* did not show any significant change in cell size following heat stress (Figure 3a,b). Similarly, an increase in palisade mesophyll cell size was also detected following heat treatment in *csn5a-1* (Appendix A). The number of abaxial cells did not change in 6th true leaf of Col-0, *csn1-10*, and *csn5a-1* following heat stress (Figure 3c), indicating that the increase in leaf size of *csn5a-1* (Figure 3d) is primarily a result of increased cell size.

Increased cell size is often a result of endo-reduplication [36]. Different colors were assigned to represent the predicted ploidy of abaxial epidermal cells (based on Reference [37]), 2n (green), 4n (orange), and 8n (red), which clearly showed the presence of larger cells (8n) in *csn5a-1* after heat treatment but not in *csn5a-1* control plants. To confirm the level of ploidy in *csn5a-1* following heat treatment, we employed flow cytometry. As anticipated, for *csn5a-1*, we observed an increase in 8n nuclei (12.5% to 22.4%) and decrease in 2n nuclei (45.8% to 31.4%) (Figure 4a,b) after heat treatment. Col-0 and *csn1-10* did not show any significant change in the ploidy level after heat stress. Thus, heat stress plays a crucial role in inducing endo-reduplication in *csn5a-1*, which we predict results in the increased cell size, increased photosynthetic capability, and increased plant size.

### 3.4. Deneddylation of CUL1, But Not CUL3, is Enhanced in csn5a-1 after Heat Treatment

To determine if the heat stress conditions influence deneddylation activity of the mutant CSN in *csn5a-1* seedlings, we employed immunoblot analysis to determine the levels of CSN5 protein and the relative neddylation state of CUL3 and CUL1. As seen in Figure 5a, the levels of CSN5 remained stable following heat shock. Similarly, we could detect no difference in neddylation state of CUL3 following the heat treatment (Figure 5b). However, we detected an increase in deneddylated CUL1 after heat stress in *csn5a-1* (Figure 5c). Densitometry analysis using ImageJ revealed a 5-fold increase of CUL1/CUL1^Nedd^ ratio in *csn5a-1* following heat treatment, whereas the WT showed a small increase of only 0.19-fold (Appendix A). Thus, the phenotypic changes detected in *csn5a-1* can possibly be explained by a rescue of CSN-dependent CUL1 deneddylation activity.

### 3.5. Auxin Activity Increases in csn5a-1 Following Heat Treatment

*csn5a-1* is defective in the auxin-response pathway [24]. Thus, to understand the role of auxin in the enhanced growth of *csn5a-1* following heat stress, we analyzed expression levels of the auxin-induced genes SAUR7 (Small auxin-up RNA 7), SAUR19, EXPA4 (Expansin A4), and EXPA10. We observed 1–3 log2 fold downregulations in these genes in *csn5a-1* seedlings compared to Col-0, confirming the reduced auxin-response in *csn5a-1*. The expression of SAUR19 and EXPA4 increases 3 h after heat treatment in *csn5a-1* seedlings but not in heat-treated Col-0 (Figure 5d).

To further test the increased activity of CSN-dependent deneddylation of CUL1 containing CRL, we assayed the activity of a known SCF target—the auxin-induced activity of SCF^TIR1^-mediated protein degradation. We crossed *csn5a-1* with the auxin reporter lines DR5::N7-VENUS and DII-VENUS. The DR5::N7-VENUS signal is under the regulation of several ARF (auxin response factor) binding sites [29,38] and thus is directly related to auxin levels, whereas in DII-VENUS, VENUS protein is attached to auxin interaction domain II of AUX/IAA [30] and thus the VENUS signal is inversely related to auxin levels. We generated two homozygous mutant lines: *csn5a-1* × DR5::N7-VENUS and *csn5a-1* × DII-VENUS. We confirmed the integration and homozygosity by PCR in the F2 population and used the seeds for further study.

Seeds were grown on liquid media and confocal imaging was done 3 h as well as 3 d after the heat treatment (2 h a day at 44 °C, for 6–13 d). We observed significant increases in the levels of VENUS signal in *csn5a-1* × DR5::N7-VENUS 3 h after the final heat treatment compared to untreated plants, while no such increase was found in the WT (DR5) background (Figure 6a). Three days following the final heat treatment, the level of DR5::N7-VENUS fluorescence reverted to control levels. However, the fluorescence level of heat treated *csn5a-1* × DR5::N7-VENUS remained high compared to the control even after three days (Figure 6b), indicating an increase in auxin activity after heat treatment in *csn5a-1*.

To determine if the increase in auxin activity is due to a decrease in auxin repressor activity, we assayed the homozygous line *csn5a-1* × DII-VENUS. In this line, higher VENUS levels indicate higher levels of the auxin repressor and lower levels of auxin activity. We observed a decrease of the DII-VENUS fluorescent level in both DII-VENUS and *csn5a-1* × DII-VENUS lines 3 h after the final heat treatment, suggesting that heat enhances the degradation of the auxin repressor (Figure 7a). Three days after the final heat treatment, DII-VENUS fluorescence levels in the control line returned to baseline values (e.g., pre-heat treatment). On the other hand, in the *csn5a-1* × DII-VENUS line, DII-VENUS fluorescence levels remained low for three days following the final heat treatment (Figure 7b), indicating enhanced degradation and maintained auxin response. These results correlate with the increased deneddylation of CUL1 in *csn5a-1* following heat stress. Thus, heat stress increases CUL1 deneddylation, which mediates degradation of AUX/IAA repressor to increase the auxin signaling and growth of *csn5a-1*.

## 4. Discussion

Numerous studies of the COP9 signalosome have highlighted its role in regulating plant responses to diverse signals [6,7,8,9,10,11,13,14,18]. Here, we showed that exposure to heat stress partially rescues the dwarf phenotype of the *csn5a-1* mutant. This rescue manifested at numerous levels. At the level of growth morphology, a phenomic analysis revealed a gradual, significant increase in rosette size following heat treatment in *csn5a-1* but not in the WT (Col-0). The increased growth detected was also seen at the cellular level, with both leaf epidermal and mesophyll cells showing increased size in *csn5a-1* following the heat treatment. This increased cell size correlated with a rescue of ploidy in epidermal cells.

Physiologically, we detected an increase of net photosynthetic CO_2_ assimilation rate following heat stress. In WT and *csn5b-1*, the net photosynthetic CO_2_ assimilation rate remained elevated for four days following the heat regimen. For *csn5a-1*, the net photosynthetic CO_2_ assimilation rate remained elevated for 11 days following heat treatment (Figure 2a). Physiologically, leaf growth is regulated by carbon availability, hydraulic properties, and wall extensibility, which promotes cell division, cell expansion, and increase in ploidy [39]. Thus, the increased CO_2_ assimilation rate in *csn5a-1* indicates an increase in photosynthesis rate and is likely the main cause for the increases in growth of the mutant.

When first confronted with the heat-enhanced growth phenotype of *csn5a-1*, we considered that this heat-enhanced growth phenotype may be a characteristic of other photomorphogenic mutants. However, it appears to be specific for the mutation in *csn5a-1,* as the growth of *csn1-10*, *csn5b-1*, *cop1-4*, and *spa1-3-4* was not enhanced following the same heat treatments.

A straightforward explanation for the improved growth of *csn5a-1* following heat is that CSN activity could be enhanced under elevated temperatures. Indeed, we detected enhanced deneddylation of CUL1 in *csn5a-1* following heat conditions (Figure 5c). In the absence of CSN5A, the deneddylation activity in this mutant suggests the contribution of CSN complexes containing CSN5B [40]. Thus, it appears that the catalytic activity of this specific subset of CSN complexes is heat enhanced. On the other hand, we could not detect a change in CUL3 deneddylation status in *csn5a-1* following the same heat conditions (Figure 5b). While CUL3 forms the E3 complex with RBX1 and BTB proteins and is involved in ethylene synthesis [41], CUL1 coordinates SCF E3 complexes together with SKP1 and RBX1 and has a major role in auxin signaling [7]. Thus, the enhanced growth of the *csn5a-1* correlates with increased deneddylation activity of CUL1, which increases auxin signaling following heat treatment.

We propose that, overall, CSN5A functions in buffering a plant’s response to heat, partially through regulating the auxin response to heat. Auxin is of course essential for plant growth, including regulation of cell division, cell expansion, and differentiation [42,43,44]. Accordingly, auxin-resistant mutants are impaired in cell division, cell expansion, and lateral root formation [45,46]. *csn5a-1* also shows similar phenotypes which implicated CSN5 and the CSN-mediated degradation of AUX/IAA in regulating auxin responses such as cell division and expansion [7].

Indeed, we found that auxin signaling is enhanced in *csn5a-1* following heat stress. In control lines, heat induces an increase in auxin signaling, which reverts to baseline levels three days following the heat treatment. In *csn5a-1,* this heat-induced increase in auxin signaling was maintained even after 3 d of stress (Figure 6b), suggesting a role of CSN5A in auxin homeostasis following recurrent heat treatment.

These results are in line with earlier transcriptomic analyses carried out in our lab which suggested that heat stress influences auxin signaling. In the short duration of 3 h of heat stress, we found a change in the auxin response pathway (Appendix A). Network analysis revealed that the heat-induced upregulated auxin pathway genes fall mostly in the GH3 and Small auxin-up RNA (SAUR) families, forming two major networks. The first network connects SAUR55, SAUR16, GH3.1, and GH3.17 while the second connects SAUR14, SAUR6, and the O-fucosyltransferase family (Appendix A). Auxin mediates acid growth by loosening the wall at low pH, which elevates wall extensibility and results in fast cell elongation [47,48], and this process is controlled in part by an additional SAUR, SAUR19 [49]. Further, this acid growth requires EXPANSINS which are non-enzymatic wall-loosening proteins which break polysaccharide networks by fragmenting and loosening the connection between cellulose microfibrils and polysaccharides [31,50,51].

We propose that recurrent heat stress leads to the degradation of the AUX/IAA auxin repressor in a CUL1-deneddylation dependent manner, resulting in higher auxin activity, and that this leads to an induction of downstream genes such as SAURs (e.g., SAUR19) and EXPANSINs (e.g., EXPA4), as was shown in the expression study in Figure 5d. These genes then function in pathways, resulting in cell elongation and plant growth. CSN5A is required for AUX/IAA recovery following heat stress to maintain proper auxin homeostasis in plants.

## Figures and Tables

**Figure 1 biomolecules-09-00805-f001:**
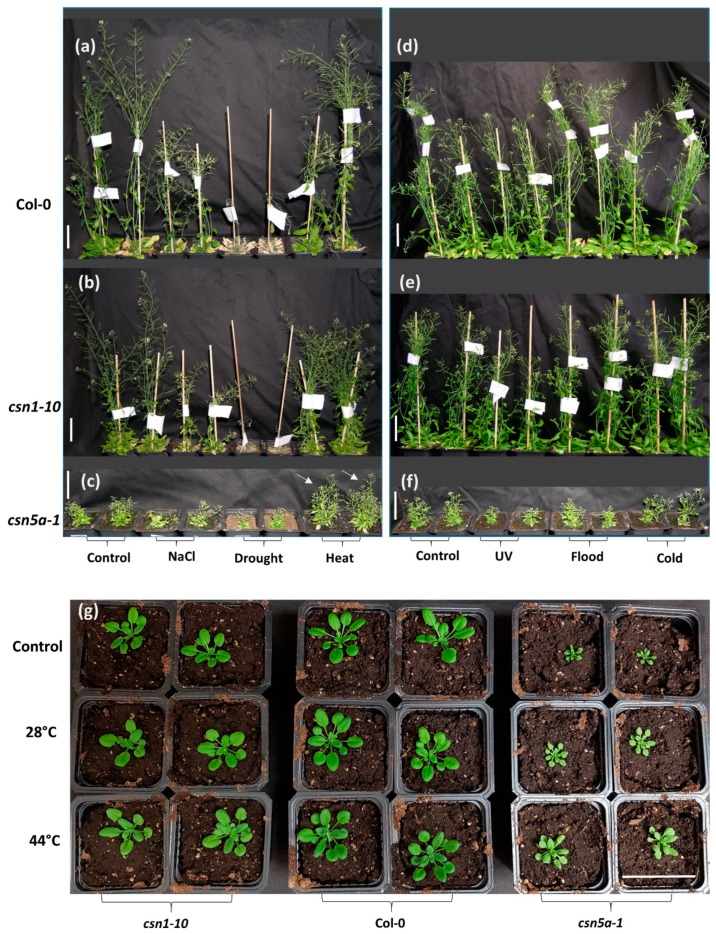
Phenotypic characterization of COP9 (constitutive photomorphogenesis 9) signalosome (CSN) hypomorphic strains (*csn5a-1* and *csn1-10*) under various environmental stress: (**a**–**c**) Phenotype of plants 70 days after sowing (DAS): Col-0, *csn1-10*, and *csn5a-1* plants respectively under salt (200 mM NaCl, 21 d), drought (21 d), and heat (2 h, 44 °C, 7 d) stress. White arrows indicate enhanced growth of *csn5a-1* under heat stress (2 h, 44 °C, 7 d) in comparison to the control condition. (**d**–**f**) Phenotype of plants 60 DAS: Col-0, *csn1-10*, and *csn5a-1* respectively under UV-C (5000 erg, 1 min), flood (7 d), and cold (4 °C overnight, 7 d) stress. (**g**) Phenotype of plants 30 DAS: *csn1-10*, Col-0, and *csn5a-1* seedlings underwent temperature stresses of 28 °C (2 h, 14–21 DAS) and 44 °C (2 h, 14–21 DAS). Scale bars, 5 cm.

**Figure 2 biomolecules-09-00805-f002:**
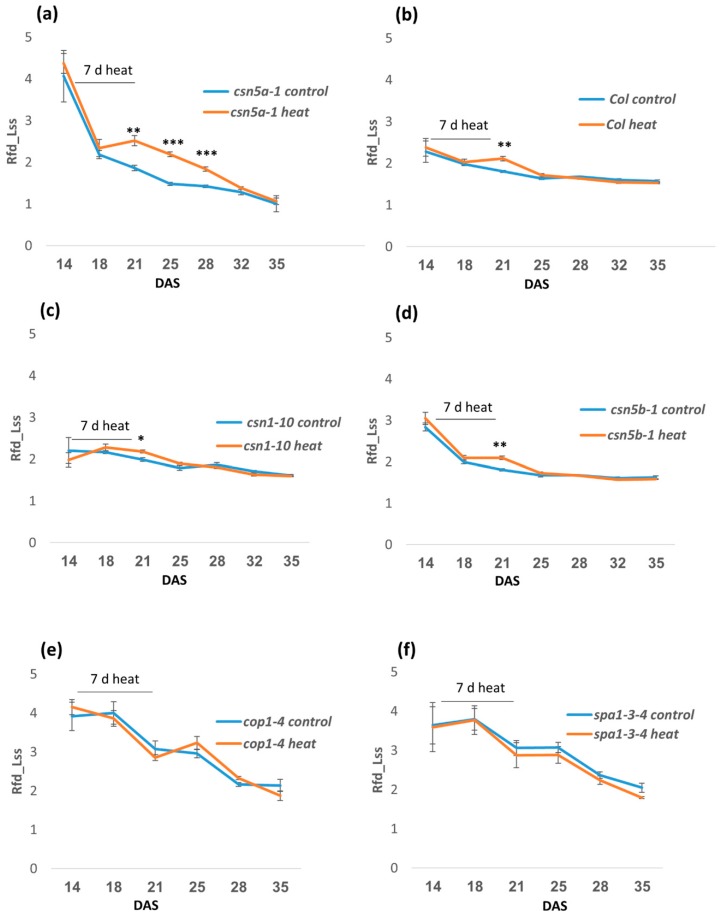
Phenomics study showing time series of relative fluorescence decline ratio in steady-state (Rfd_Lss) for the photo-morphogenesis repressor mutants following heat stress (2 h, 44 °C, 14–21 DAS). (**a**) Rfd_Lss increases significantly in *csn5a-1* after 7 d heat stress and continues to be higher for 11D before getting equal to *csn5a-1* control. (**b**) Rfd_Lss increases in Col-0 after 7 d heat stress but becomes equal to Col-0 control after 4D. (**c**) Rfd_Lss increases in *csn1-10* after 7 d heat stress but becomes equal to *csn1-10* control after 4D. (**d**) Rfd_Lss increases in *csn5b-1* after 7 d heat stress but becomes equal to *csn5b-1* control after 4D. (**e**) Rfd_Lss does not change in *cop1-4* after 7 d heat stress. (**f**) Rfd_Lss does not change in *cop1-4* after 7 d heat stress. Error bars represent SEM of biological replicates (*n* = 4). Student’s t test * *p* < 0.05; ** *p* < 0.01; *** *p* < 0.001.

**Figure 3 biomolecules-09-00805-f003:**
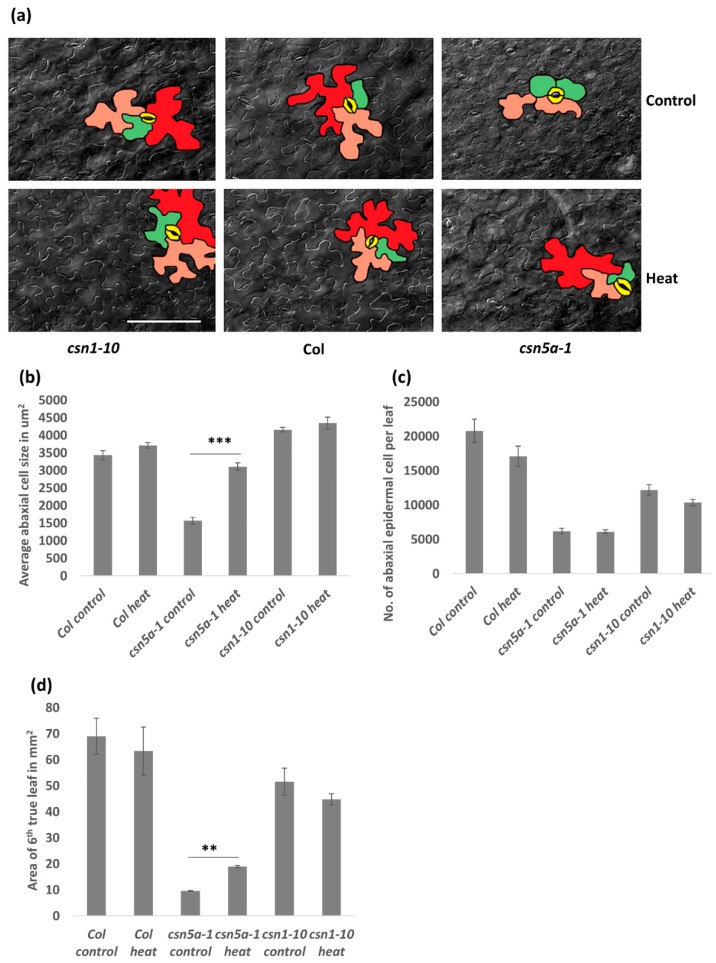
Image and graph of abaxial epidermal cell size in 6th true leaf of plants 30 DAS having undergone 7 d heat treatment: Cell size increases in *csn5a-1* after 7 d heat treatment. (**a**) Differential interference contrast (DIC) image of abaxial epidermal cell after 7 d heat treatment of 30 DAS *csn1-10* (left), Col-0 (middle), and *csn5a-1* (right). Cells colored in different color are most likely representing, 2n (green), 4n (orange), 8n (red), and stomata (yellow). Scale bar, 100 µm. (**b**) Graph displaying average abaxial cell size in µm^2^. (**c**) Graph displaying the number of abaxial cells per 6th true leaf. (**d**) Graph displaying area of 6th true leaf. Error bars represent SEM of biological replicates (*n* = 2–4). Student’s t test ** *p* < 0.01; *** *p* < 0.001.

**Figure 4 biomolecules-09-00805-f004:**
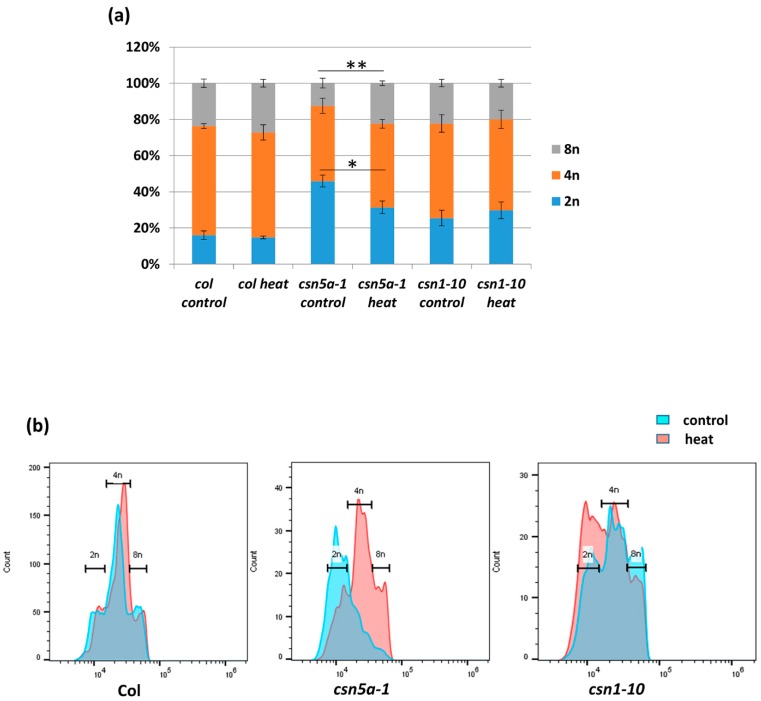
Difference in the percentage of nuclear ploidy after 7 d heat treatment in *csn5a-1* 30 DAS. (**a**) Percentage of diploid, tetraploid, and octaploid nuclei in 30 DAS Col-0, *csn5a-1*, and csn1-10 seedlings following 7 d heat treatment showing increase in the ploidy of *csn5a-1* following heat treatment. (**b**) Graph showing increase in the level of higher ploidy (4n and 8n) of *csn5a-1* (middle) following heat treatment whereas Col-0 (left) and *csn1-10* (right) does not show increase in higher ploidy (4n and 8n). Error bars represent SEM of biological replicates (*n* = 3). Student’s t test * *p* < 0.05; ** *p* < 0.01.

**Figure 5 biomolecules-09-00805-f005:**
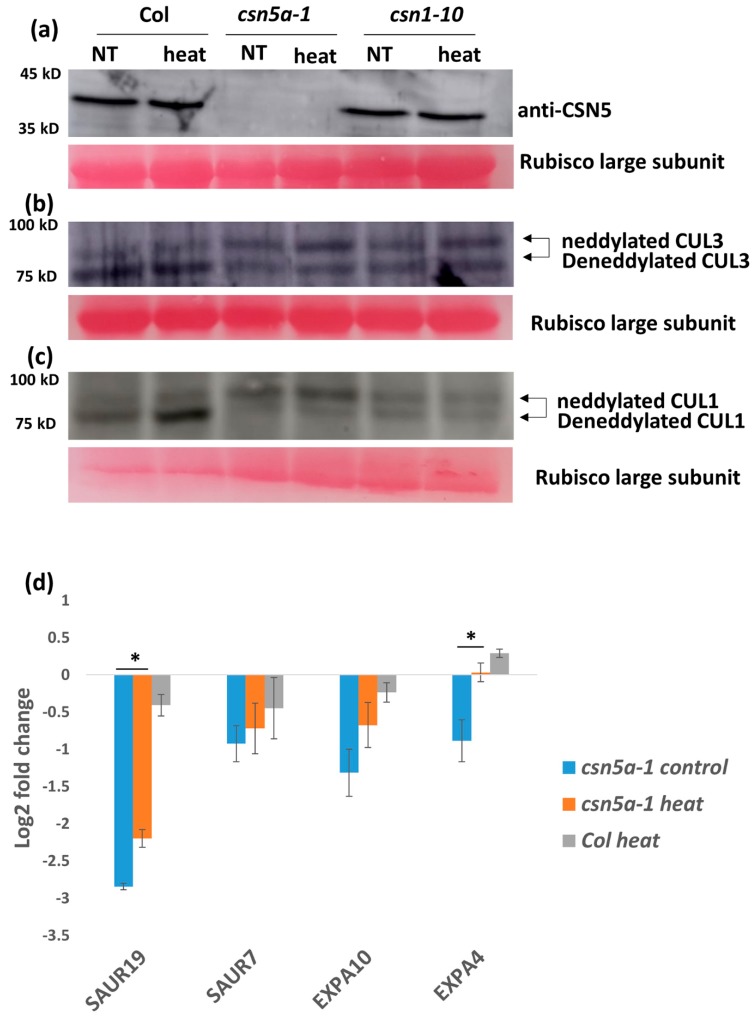
Increased growth of *csn5a-1* following 7 d heat treatment is due to an increase in deneddylation activity (against CUL1) and auxin response. (**a**) Immunoblot using equal protein concentrations with CSN5 antibodies did not show any change in the protein band of *csn5a-1* after heat treatment. (**b**) Immunoblot using equal protein concentrations with CUL3 antibodies did not show any change in cullin neddylation/deneddylation (93/85 kDa) ratio in either in *csn5a-1* or Col-0 after heat treatment. (**c**) Immunoblot using equal protein concentrations with CUL1 showed increase in deneddylation activity of *csn5a-1* and Col-0. Rubisco large subunit stained with ponceau is used as loading control. NT, non-treated. (**d**) Expression of auxin responsive genes (SAUR19 (Small auxin-up RNA 19) and EXPA4 (Expansin A4) of which the expression was downregulated in *csn5a-1* control plants compared to WT (Col-0) control plants increases after heat treatment. Expression of WT (Col-0) control plant is taken as the baseline (0) in the log2 fold change. Error bars represent SEM of biological replicates (*n* = 3–5). Student’s t test * *p* < 0.05; ** *p* < 0.01.

**Figure 6 biomolecules-09-00805-f006:**
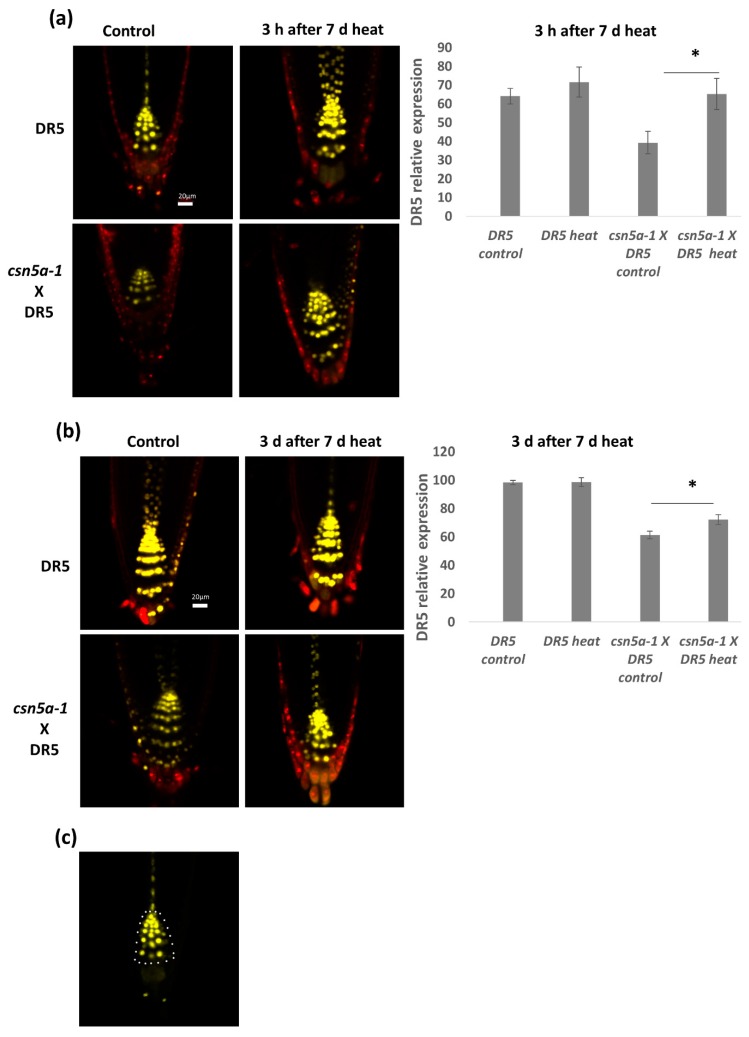
Auxin activity increases in *csn5a-1* following 7 d heat treatment: (**a**) Relative integrated density in the roots of DR5::N7-VENUS and *csn5a-1* × DR5::N7-VENUS, 3 h after 7 d heat treatment. (**b**) Relative integrated density of DR5::N7-VENUS and *csn5a-1* × DR5::N7-VENUS, 3 d after 7 d heat treatment. Cell wall was stained with propidium iodide. (**c**) White dots represent the area used to quantify DR5 expression levels. Scale bars, 20 µm. Error bars represent SEM of biological replicates (*n* = 4–7). Student’s t test * *p* < 0.05.

**Figure 7 biomolecules-09-00805-f007:**
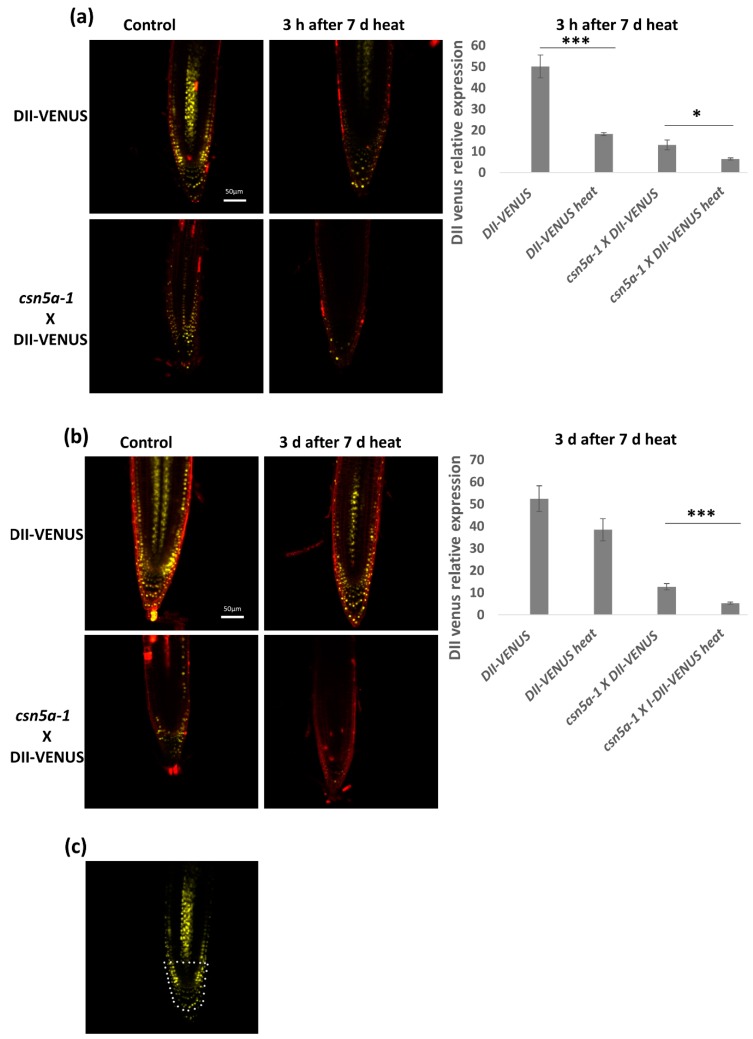
Auxin repressor activity remains low even after 3 d of 7 d heat treatment. (**a**) Relative integrated density in the roots of DII-VENUS and *csn5a-1* × DII-VENUS, 3 h after 7 d heat treatment. (**b**) Relative integrated density of DII-VENUS and *csn5a-1* × DII-VENUS, 3 days after 7 d heat treatment. Cell wall was stained with propidium iodide. (**c**) White dots represent the area used to quantify VENUS expression levels. Scale bars, 50 µm. Error bars represent SEM of biological replicates (4–7). Student’s t test * *p* < 0.05; *** *p* < 0.001.

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
