# Peer review of "CSN5A Subunit of COP9 Signalosome Temporally Buffers Response to Heat in Arabidopsis"

_biomolecules, 2019, doi:10.3390/biom9120805_

Round 1

Reviewer 1 Report

The authors have studied the roles of the COP9 signalosome (CSN) during environmental stress, particularly heat stress, using hypomorphic mutants of Arabidopsis. The authors have tested various parameters of the mutants and concluded that CSN5A has a specific role in deneddylation of CUL1 and that CSN5A is required for the recovery of AUX/IAA repressor levels following recurrent heat stress to regulate auxin homeostasis in Arabidopsis.

After careful review of this manuscript, I strongly believe that the authors have provided sufficient background, explained very well the established methodologies, presented the results well, and concluded appropriately based on the data available. However, I do have a few major issues with this manuscript’s presentation.  Specifically,

In Abstract, please indicate at the beginning that Arabidopsis is being used in this study. In the Results, many sentences are repeated from the M&M and should be omitted. My major concerns are about the Discussion, which did not really discuss the results at all. The authors have cited only 6 papers in two paragraphs but without any serious discussion of the results from these studies at all, making the conclusions withdrawn weakly supported by the results in this study. There are quite some editorial errors that need to be corrected. For example, there should be a space between the number and its unit, 200 g but not 200g.

Author Response

Response to Reviewer 1 Comments

Reviewer's comment 1: In Abstract, please indicate at the beginning that Arabidopsis is being used in this study.

Our response: According to the reviewer's suggestion we incorporated Arabidopsis at the beginning of the abstract and the sentence changed as "Using Arabidopsis as model organism, we used CSN hypomorphic mutants to study the role of the CSN in plant responses to environmental stress, and found that heat stress specifically enhanced the growth of csn5a-1, but not the growth of other hypomorphic photomorphogenesis mutants tested".

Reviewer's comment 2: In the Results, many sentences are repeated from the M&M and should be omitted.

Our response: The repeated sentences have been omitted from the result section.

Reviewer's comment 3: My major concerns are about the Discussion, which did not really discuss the results at all. The authors have cited only 6 papers in two paragraphs but without any serious discussion of the results from these studies at all, making the conclusions withdrawn weakly supported by the results in this study.

Our response: According to the suggestion, the papers cited in the discussion are being discussed thoroughly in the revised manuscript with the addition of a few more references in the track change mode.

Reviewer's comment 4: There are quite some editorial errors that need to be corrected. For example, there should be a space between the number and its unit, 200 g but not 200g.

Our response: Space has been provided between the numbers and their units in the revised manuscript in track change mode.

Reviewer 2 Report

In the manuscript submitted to Biomolecules (*) authors works on the CSN5A subunit of COP9 signalosome temporarily buffers response to heat in Arabidopsis. This reviewer suggest the publication in Biomolecules after minor revision.

Theme is interesting, techniques, instrumentation and apparatus, stress treatments, phenomics measurements, and other analysis, are the adequated to solve this kind of analysis. Application and results are very interesting.

Minor comments:
* In Experimental, for Instrumentation, Materials and Reagents, or Programs and Databases (as SPSS, Excel, and others) ever, Product (Manufacturer, City, Country), in this order and format. Please correct in some places. In the case of USA products: Product (Manufacturer, City, State, USA).
* The number of Figures must be reduced and some of them could be provided as suplementary material.

Author Response

Response to Reviewer 2 Comments

Reviewer's comment 1: In Experimental, for Instrumentation, Materials and Reagents, or Programs and Databases (as SPSS, Excel, and others) ever, Product (Manufacturer, City, Country), in this order and format. Please correct in some places. In the case of USA products: Product (Manufacturer, City, State, USA).

Our response: The correction has been done in the material and method section according to the suggestion.

Reviewer's comment 2: The number of Figures must be reduced and some of them could be provided as suplementary material.

Our response: We have moved Figure 2 in the supplementary material and changed the figure number accordingly in the revised manuscript.

Round 2

Reviewer 1 Report

I appreciate very much the efforts that the authors have devoted to the improvement of this manuscript. I have no further comments on this manuscript.